# Potato By-Products as a Source of Natural Chlorogenic Acids and Phenolic Compounds: Extraction, Characterization, and Antioxidant Capacity

**DOI:** 10.3390/molecules26010177

**Published:** 2020-12-31

**Authors:** Nicolas Joly, Kaies Souidi, David Depraetere, Daniel Wils, Patrick Martin

**Affiliations:** 1Unité Transformations & Agroressources, Université Artois, UniLaSalle, ULR7519, F-62408 Béthune, France; nicolas.joly@univ-artois.fr; 2IUT Béthune, Université Artois, F-62408 Béthune, France; kaies.souidi@gmail.com (K.S.); david.depraetere@univ-artois.fr (D.D.); 3Nutrition & Health R&D, Roquette, F-62136 Lestrem, France; daniel.wils@roquette.com

**Keywords:** potato, phenolic, chlorogenic

## Abstract

Total phenolic compounds (TPC) and the chlorogenic acids content of potato by-product extracts of two hydro alcoholic solvents (methanol, ethanol) and two extraction methods (maceration and heating-assisted extraction) were studied. The content of TPC in the extracts was determined spectrometrically according to the Folin–Ciocalteu procedure and calculated as chlorogenic acid equivalents. Soluble phenolic acids, especially the chlorogenic acids, were performed by HPLC. The antioxidant activity of potato by-product extracts was determined by using the total oxygen radical absorbance capacity (ORAC) method. The highest content of TPC was found in raw and lyophilized red waters when using ethanol as a solvent around 57 mg/g fresh weight. Heating-assisted extraction enhances this quantitative increasing. At the given operating conditions, unpeeled potato samples exhibit a higher TPC than peeled ones, showing that TPC are accumulated in skin tissue. The greatest amount of chlorogenic acid (Caffeoyl-Quinic Acids, 3, 4, 5 CQA), mainly the 5-CQA (870 ± 39.66 µg/g WM for wet matter versus DM dry matter), was obtained in the pellets and lyophilized fresh peels (skin vs. flesh). In addition, the greatest amounts of chlorogenic acids were found when potato peels were extracted with methanol. Heating-assisted extraction improved the chlorogenic acid concentration of the potato peel extracts. The total ORAC amounts recorded in the different potato fractions varied between 1500 and 1650 µM TE/g. They were higher than those of some fruits, vegetables, nuts, cereals, and sweet potato cultivar. The good correlation coefficient found between TPC, chlorogenic acids determination, and the ORAC capacity indicates that the TPC can be used as a good indicator of the antioxidant capacity of potato by-products.

## 1. Introduction

Vegetal raw material processing generates large quantities of co-products (or by-products), which are affordable, and valuable starting material for the extraction of value-added compounds such as dietary fibers, natural antioxidants, biopolymers, and natural food additives [1,2,3]. In addition to co-products of the milling, oil, dairy, sugar industries, and other food manufacturers, the potato starch industry by-products are of great interest.

In several countries, potatoes botanically called *Solanum tuberosum* L. are one of the most important basic crops for human consumption, together with wheat, rice, and corn. According to the Food and Agriculture Organization (FAO) [4], China is the biggest potato producer with an output of 72,000,000 tons, followed by Russia (35,718,000 tons) and India (26,280,000 tons). However, France is producing 6,271,000 tons annually.

By-products derived from potato processing can be divided into two major categories: cull or discarded potatoes (whole or cut potatoes not destined for human consumption), and potato processing wastes (derived from the manufacture of potato ingredients or potato-based food products) [5]. Both discarded tubers and potato by-products represent a disposal problem to the potato industry, since the wet wastes constitute a source of plant spoilage and pathogenic infections [6]. Furthermore, these co-products are either used as cattle feed or as a source for biofuel production or biotechnological a substrate for microbial and enzymatic fermentation or cell production.

On the other hand, and taking into account the growing rejection of synthetic food additives by consumers, functional ingredients obtained from natural sources may be a promising alternative. The utilization of by-products also contributes to reducing the amounts of waste and thus is a way of participating in more sustainable production [6].

In the potato industry, peels, which are the major portion of processing wastes, contain a spectrum of nutritionally and pharmacologically interesting components such as phenolic compounds (chlorogenic acids, ascorbic acid, flavonoids, etc.), glycoalkaloids [7,8,9], cell wall polysaccharides, and dietary fiber [10,11]. Moreover, Mukherjee et al. [12] used potato peel as a low-cost agro-industrial medium in the production of both alpha-amylase and alkaline protease enzymes, respectively, to be used as detergents.

Potato peels have been shown to contain several important and high added-value phenolic antioxidants, which may be used as natural antioxidants that can substitute for synthetic ones.

Friedman [13] reported that potato peels contain a high content of polyphenols, which was reported to be 10 times higher than their levels in the flesh accounting for approximately 50% of all polyphenols in potato tuber. In potato peels, the largest part of phenolic acids consists of chlorogenic acids (Caffeoyl-Quinic Acids, 3, 4, 5 CQA), which reached up to 90% of phenolics [5,6,13,14]. Mattila and Hellstrom [15] reported that caffeic acid derivatives (chlorogenic acid) as the main phenolic constituents in potatoes. The phenolics are present in both the skin and the potato flesh, the concentration being higher in the skin. Lewis et al. [16] reported that the fresh pulp and skin of potatoes contain 30 to 900 mg/kg and 1000 to 4000 mg/kg, respectively of chlorogenic acid and minor amounts of other phenolic acids between 0 and 30 mg/kg.

Rodriguez De Sotillo et al. [17] measured both amounts and type of phenolic compounds in freeze-dried, aqueous extract of potato peels. The authors reported that chlorogenic (50.31%), gallic (41.67%), protocatechuic (7.815%), and caffeic (0.21%) acids were the major phenolic compounds detected in their study. Moreover, this aqueous freeze-dried extract proved to be stable during 2 weeks of storage and was as effective as butylated hydroxyanisole (BHA) in inhibiting the lipid oxidation of sunflower oil.

Furthermore, aqueous extracts from potato peel by-products also showed free radical scavenging activities in various in vitro assays [18]. Mansour and Khalil [19] found that 90% ethanol extracts of potato peels exhibited antioxidative activity in model systems (β-carotene/linoleic acid emulsions) and ground beef patties.

Several researchers have employed conventional solid–liquid extraction technics, such as soxhlet, heat reflux, and maceration [20,21,22,23,24] to extract phenolic antioxidants from potato peels and have estimated the antioxidant activity of the extracts.

Recently novel extraction techniques such as microwave-assisted extraction (MAE), ultrasound-assisted extraction (UAE) and supercritical fluid (SF) have been widely used in the extraction of phenolics from agro-industrial waste samples. Singh et al. [23] have studied the MAE process for the extraction of antioxidants from potato peels. Chen et al. [24] have used MAE to extract solanesol from potato leaves and stems. Almost all the researchers reported that the MAE process provided significantly higher yield than conventional methods of extraction. Microwave heating also allows a reduction in the volume of solvent and sample and time needed for the extraction, thus being a high-cost effective solution.

Recently, researchers have also shown the impact of the method of production of potatoes, conventional or not (organic, biodynamic) on the level of bioactive compounds [25,26].

The present work was undertaken in order to examine the potential interest of potato processing co-products supplied by a local French potato industry according to the following steps: (i) Study the extraction of phenolics (total phenolic content) and more specially (ii) the chlorogenic acids from starch potato peel, potato flesh, liquid pulp, lyophilized one and red waters using alcohol solvents and different extraction methods and (iii) Evaluate the antioxidant activity using the oxygen radical absorbance capacity (ORAC) test.

## 2. Results–Discussion

### 2.1. Choice of the Method of Extraction

Table 1 sums up the results of total phenolic compounds (TPC) expressed as milligrams of chlorogenic acid equivalent per g of sample and chlorogenic acid contents obtained on ground sieved potato pellets (GSP) by the application of different extraction methodologies using methanol as a solvent.

We observed that there is a concentration variability of these compounds using a methanol-based extraction process. The TPC in the GSP potato sample in our experiments varied in a wide range from 1.53 ± 0.18 mg/g (percolation method) to 4.78 ± 0.23 mg/g (heating method at 40 °C/30 min) (Table 1). In comparison, in the case of chlorogenic acid extraction, heating showed the best process (106.45 ± 3.55 µg/g) followed by maceration (102.64 ± 7.06 µg/g) and ultrasound (94.84 ± 5.40) µg/g. Taking into consideration these results, we have further only focused on heating and maceration techniques for the following parts of the work.

### 2.2. Total Phenolic Compounds (TPC) in Potato By-Products

The antioxidant activity is usually attributable to the polyphenols, mainly phenolic acids and flavonoids, which are the most important sub-families of phenolic compounds found in fruits and vegetables.

In many cases, TPC are expressed as chlorogenic acid equivalents. This is usual in analytic research, since the free soluble phenols are determined using methanol, ethanol, and acetone solvents. Therefore, in our case, the concentration of TPC in the methanol and ethanol extracts, expressed as mg CQA equivalent/g sample, in fresh weight (fw) or dry weight (dw) were dependent on both the type of potato co-products and extraction method used, as shown in Figure 1.

Figure 1 shows that the results of TPC are dependent on the type of potato by-products whatever the extraction method and the type of solvent.

We observe that for different samples of potato, the TPC content on the methanolic extract remained lower than 10 mg/g (Figure 1a). The red liquid sample (RL) exhibits a higher amount of TPC of about 7.7 ± 0.21 mg/g when using the maceration method. This amount is approximately seven times greater than the value obtained by application of the heating method. The total phenolic content in the GSP sample was 4.78 ± 0.23 mg/g, which was obtained with heating extraction, and it is two times higher than that obtained with the maceration technique. The heat processing of methanol extract of lyophilized fresh potato peels (LFPPe) allowed measuring 3.83 ± 0.13 mg/g of TPC in comparison to extraction by the maceration method, which gave an amount of 2.99 ± 0.17 mg/g. Then, we can conclude that lyophilized flesh potato (LFPo) and dehydrated potato pulp (DPP) exhibit approximately the same content of TPC of about 1.5 mg/g. It appears that when using MeOH: H_2_O as a solvent, the TPC contents in the different potato samples can be sorted in the following order: RL > GSP > LFPPe > DPP > LFPo > LGSP.

When compared to published data, Al-Weshahy and Venket Rao [22] reported that the total phenolic content ranged from 1.51 to 3.32 mg of gallic acid equivalent per gram of dry potato peel powder. Peel samples of the two varieties of potato had a level of TP ranged from 1.1 to 2.4 mg/g dw [27]. Samarin et al. [28] reported that the amount of phenolic compounds in the methanol extract of potato peels was around 522.1 ± 2.14 μg GAE/g dw.

From Figure 1b, we can observe that the highest concentration of TPC was found in the ethanol extract for both RL (56.9 ± 2.04 mg/g) and LRL (46.2 ± 12.05 mg/g) samples. These high values were obtained with the heating extraction and are 10 times greater than those found when using maceration methodology. Furthermore, the amount of TPC compounds in the ethanol extract was higher than in the methanol extraction process and TPC concentrations in the potato by-products were RL > LRL > GSP > LFPPe > LFPo > DPP.

Figure 1b also shows that heating allowed a better extraction of the TPC on RL and LRL in aqueous ethanol compared to the corresponding samples obtained with methanol. Furthermore, RL and LRL exhibit a higher level than the other potato coproducts. This may be related to amount of anthocyanin and pigment, which confer the red color to this product. On the other hand, the Folin–Ciocalteu reagent can overestimate the total phenolic content, since it also reacts with hydroxyl groups in amino acids and sugars in addition to phenolic groups [29].

According to the literature, the increased levels of polyphenol compounds may also be related to the increased production of reducing sugars during heating, which are known to interfere with the Folin reagent ([30,31]).

Based on our results, it seems that there are no significant differences among the types of by-products. We also observe in the ethanol extract under the heating and maceration process that peels (in pellet form (GSP) or fresh) had TPC contents that were slightly greater than the values in the flesh (lyophilized or not) or the whole potato (CPo). The differences in TPC content obtained are probably due to the color, nature, and variety of potatoes.

Our results for TPC extraction are similar to or higher than those reported in the literature. Therefore, Truong et al. [32] were interested in determining the phenolic composition in the ethanol extract of sweet potato leaves and roots. The authors found that the peel samples from the three cultivars had total phenolic contents from 145.1 to 181.7 mg CQA equivalents/100 g fw, which were over two to three times greater than the values in the flesh (57.1 to 78.6 mg/100 g fw) or the whole roots. Teow et al. [33] reported that the TPC contents were in the range of 3.3 to 94.9 mg/100 g fw in the sweet potato cultivars. However, the phenolic content of 945 mg/100g fw was found in red-fleshed sweet potato roots [34].

In all cases, the TPC in the unpeeled potato samples was higher than in the peeled samples, showing clearly that TP is accumulated in the skin tissue. This is in agreement with literature data [35] and [36]. For example, TPC varied in the peeled potato samples from 191 to 1864 mg/100 g dry matter (DM) mean, while these parameters varied from 345 to 2852 mg/100 g DM in unpeeled samples. The TPC of the free fraction in red or purple potatoes, expressed in gallic acid equivalent (GAE), was 51.36–73.60 mg GAE/100 g, while it was 8.77–19.91mg GAE/100 g in yellow or white potatoes [37]. Albishi et al. [38] reported that the TPC in four potato varieties, including yellow and purple potatoes, ranged from 45 to 118 mg GAE/100 g. These values were lower than those obtained in our experiments with respect to the operating conditions.

The variation in TPC obtained in our experiments and those reported in the literature may be attributed to the diverse potato genotypes, growing, and culture conditions and extraction process.

### 2.3. Individual Polyphenolic Compounds (Chlorogenic Acids)

Typical gradient HPLC chromatograms of the chlorogenic standards (A) acids and in GSP fraction (B) from potato co-products are shown respectively in Figure 2.

Based on the HPLC elution times (Figure 2a), the standard compounds are identified as 3-CQA or chlorogenic acid (peak at approximately 10 min), 5-CQA (peak at 19 min), and 4-CQA (peak at 22 min).

Typical HPLC phenolic profiles of the ground and sieved pellets (GSP) extracts eluted with the gradient condition are shown in Figure 2b. Based on these figures, we can conclude that the chlorogenic derivative acids are present in the potato co-products. These compounds were respectively identified as 3-CQA (peak 2), 5-CQA (peak 4), and 4-CQA (peak 5), which were separated in the same pattern as shown in Figure 2a for the chlorogenic standards.

The HPLC results of potato samples obtained from methanol and ethanol extracts under maceration and heating techniques are shown respectively in Figure 3a,b.

We can observe that the potato samples exhibit the same trends of chlorogenic acids concentration when using both maceration and heating process. However, the concentrations of the chlorogenic acids differ depending on the solvent of extraction. Regardless methanol or ethanol solvent, chlorogenic acids were present at the highest concentration on the “LFPPe” fraction followed by the “GSP” and “LFPo” fractions in all samples.

For example, in the methanol extract (Figure 3a), when using the heating process, LFPPe recorded an average amount of 3-CQA of about 870.54 ± 39.66 µg/g fw compared to that in ethanol extract (Figure 3b), which showed a 786.1 ± 30.67 µg/g. This is common in the GSP and LFPo fractions.

The highest content of TCP was obtained with ethanol as solvent, while the highest content of chlorogenic acid obtained with MeOH can be explained by the selectivity of the extraction. Indeed, MeOH allows a better and selective extraction of CQA, while EtOH allows a more global extraction of polyphenols. Moreover, (i) highly polar solvents, such as methanol, have a high effectiveness on the extraction of antioxidant molecules, which improve the ability to penetrate the cells and obtain a higher extraction yield; (ii) in the fractionation of biomass, an EtOH/H_2_O mixture, for better efficiency, always carries out an extraction of tannins and other extractables.

Moreover, scientists have discovered and reported that highly polar solvents, such as methanol, have a high effectiveness on the extraction of antioxidant molecules, which improve the ability to penetrate the cells and obtain a higher extraction yield.

In the major potato by-products, the amount of chlorogenic acid in the maceration method was lower than those of the respective compounds obtained with the heating process. It seems that heat enhances the yield of extraction and the increases of chlorogenic concentration. The changes in the concentration of these compounds can be due to their heat-induced isomerization by heat and polyphenoloxidase action during the heating process, as reported by [32,39]. As shown in Figure 3, regardless of the operating conditions, relative concentrations on the chlorogenic acid according to different potato samples were in the following order: LFPPe > GSP > LFPo > LGSP > CPo.

It appears that chlorogenic acid is among the phenolic free forms found in potatoes, and it reported that these compounds predominate over the bound form in both skin tissue and flesh. These suggestions are in concordance with literature data [20,21,22,27,32,35,37,40].

Truong et al. [32] reported that chlorogenic acid was the predominant phenolic compound among compounds extracted from sweet potato roots of several cultivars. As a function of genotype potato cultivars, the authors found that the chlorogenic acid concentration varied from 27.8 ± 0.7 to 42.8 ± 6.6 mg/100 g fw for peel raw, from 5.1 ± 0.5 to 9.3 ± 0.9 mg/100 g fw for flesh raw, and 4.6 ± 1.1 to 13.6 ± 3.3 for whole raw. They also reported that the hot extraction of different fractions (peel, flesh, or whole) of sweet potato roots favors obtaining a high content of chlorogenic acids.

According to Ru et al. [37], the content of chlorogenic acid ranged from 2.06 to 79.91 mg/100 g DM, which was the highest in most of the potato samples. Their results seem to be relatively close to those obtained in our investigation. For example, Akyol et al. [20] reported in their review that chlorogenic acid accounted for the wide range of concentration from 0.2 to 2193 mg/100 g dry extract. These considerable variabilities were linked to genotype, culture, and the processing of potatoes.

In order to characterize their potential biological activity, these extracts were subjected to the evaluation of their antioxidant activity by total ORAC tests.

### 2.4. Oxygen Radical Absorbance Capacity (ORAC) of Potato Samples

In our study, we measured the total ORAC (hydrophilic + lipophilic) values, which were reported as µM Trolox equivalents per gram of samples in fresh weight (Figure 4a,b). These values were determined from the regression equation of the standard curve (Net Area Under the Curves (AUC) = a × [TE] + b).

From Figure 4a,b, whatever the operating conditions (nature of solvent or extraction process), the total ORAC values recorded in our study in the different potato fractions (of both peels or flesh or liquid) are compared to those of standard chlorogenic acids, i.e., 3-CQA, 4-CQA, and 5-CQA. These three chlorogenic acids showed an ORAC value of 1568.02, 1546.82, and 1593.10 µM TE/g for 3-CQA, 4-CQA, and 5-CQA, respectively. These results indicate that the different potato fractions follow the same trends and contain a large amount of chlorogenic acids.

We observe that the type of solvent has no significant effect on the ORAC values, because all co-products exhibit ORAC values similar to those of standards. It appears also that the red waters and its freeze-dried form exhibit a relatively higher ORAC values followed by the freeze-dried peels (LFPPe), the pellets (GSP, LGSP), and freeze-dried flesh (LFPo).

This indicates that freeze-drying (or lyophilization) increased the antioxidant activity of different samples as measured by ORAC tests. This might be due to the release of the compounds from the plant cells during the freeze-drying process. In addition, Pérez-Gregorio et al. [41] reported that onions subjected to freezing showed a reduced level of flavonols, whereas freeze-dried samples showed higher amount. The authors attributed this increment to the possibility of liberation of phenolic compounds from onion during freeze-drying.

The total ORAC values recorded in the different fractions (from both peels or flesh or liquid) of potatoes in our study (Figure 4a,b) were much higher than those of some fruits (apples, apricot, avocado, cherries, grapefruit, orange, pears), vegetables (broccoli, cabbages, lettuces), nuts (almond, cashew, peanuts, etc.) and sweet potato cultivars (white, yellow, purple fleshed) (Wu et al. [42]). The authors analyzed the antioxidant activity of a sweet potato cultivar based on the hydrophilic and lipophilic ORAC procedures. They found the total antioxidant activity to be 9.02 µmol Trolox equivalents (TE)/g fresh weight (fw).

Antioxidant activities of orange-fleshed sweet potato are assessed as a function of cooking process types (Kourouma et al. [43]). The authors reported that in comparison to the untreated one, the boiling potato sample exhibited a significant high ORAC index (6.31 µmol TE/g) followed by steaming treatment (5.83 µmol TE/g) and then microwaving treatment 5.56 µmol TE/g). The literature data mentioned that the certain compounds generated during cooking of potato samples contribute to increasing the global antioxidant activities [43,44]. On the contrary, Tang et al. [45] reported that the cooking process could decrease the antioxidant activity by losing the bioactive molecules.

According to Teow et al. [33], the total antioxidant activity (hydrophilic + lipophilic ORAC) was highest (27.2 µmol TE/g fresh weight (fw)) for a purple-fleshed and lowest (2.72 µmol TE/g fw) for a white-fleshed genotype potato.

Hu et al. [46] reported that potato cultivars with pigmented flesh color had significantly higher antioxidant activity than commonly consumed white or yellow-fleshed potatoes; ORAC values ranged from 42 to 168 µmol TE/g DW.

Furthermore, our results were in in good agreement with data reported by Brown et al. [47]. In Brown’s work, the hydrophilic oxygen radical absorbance capacity (ORAC) ranged from 333 to 1408 μM Trolox equivalents/100g FW, and the lipophilic ORAC ranged from 4.7 to 30 nM tocopherol equivalents/100 g fw.

Moreover, our results were compared with spices, and some of them had an extremely high total ORAC. A distinguishing feature of this group of foods is that the total ORAC values of some samples, i.e., turmeric, oregano leaf dried, cinnamon, cloves reached 1592.77, 2001.29, 2675.36, and 3144.46 µmol of TE/g, respectively (Wu et al. [42]). These results indicate that the essential oils in them contained considerable amounts of antioxidants and the major hydrophilic antioxidants in spices are derivatives of phenolic or cinnamonic acid.

Therefore, the total antioxidant activity ORAC of our potato fractions are higher than those of sorghum clones, which were in the range of 271–878 µmol TE/g dw (Awika et al. [46]). Bao et al. [47] reported a similar trend for Chinese bayberries in which black bayberry cultivars had much higher antioxidant activity than pink and yellowish varieties.

The authors reported that the constituents responsible for the hydrophilic antioxidant activity are primarily phenolic compounds and anthocyanins, whereas carotenoids and tocopherols are the main antioxidant constituents in lipophilic extracts.

The antioxidant activity data obtained from the total ORAC procedure were correlated with the total phenolic contents (Figure 5a) and chlorogenic acid (Figure 5b).

We observe that the TPC content is correlated very well with the ORAC values (about R^2^ = 0.83) whatever the type of solvent. However, the correlation between ORAC values and chlorogenic acid (5-CQA) content were pretty good when using MeOH as solvent (R^2^ = 0.86) and less high (R^2^ = 0.75) in the case of EtOH. These results show that the high degree of correlation between both TPC and chlorogenic acid analysis and the antioxidant activity of the potato extracts indicates that the assay for total phenolic would be a useful technique for the rapid evaluation of antioxidant activity in potato co-products. Based on our experimental results, we can conclude that the total phenolic molecules and particularly the chlorogenic acids present in different potato fractions were correlated with antioxidant activity results (ORAC). This positive correlation indicates that an ORAC method is the most appropriate assay to determine the antioxidant activity of TPC, especially chlorogenic acids extracted by the hydroalcoholic maceration method.

According to the literature, our correlation results seem to be comparable to the results obtained by Teow et al. [33]. The authors reported a correlation coefficient R^2^ of about 0.937 between TPC and hydrophilic ORAC for sweet potato. High correlation coefficients between phenolic content and antioxidant activities have also been reported for various food commodities such as sorghum (R^2^ = 0.971) and cactus pear (R^2^ = 0.970–0.990) by Rabah et al. [48] and Stintzing et al. [49].

Then, we can conclude that this type of assay meaning total phenolic components determination can be used as a good predictor of the global antioxidant activity of fruits and vegetables, including potatoes.

Kourouma et al. [43] reported the relationship between antioxidant activity and the content of α-carotene and β-carotene of orange fleshed sweet potato. The authors showed a different trend for ORAC and DPPH radical scavenging activity, which could be attributed to the solubility of caretonoid that depends on the solvent used.

## 3. Materials and Methods

### 3.1. Reagents

Ethanol, methanol (HPLC grade), Trolox^®^, and Folin–Ciocalteu’s Phenol Reagent 2N were purchased by Sigma-Aldrich (Saint-Quentin Fallavier, France). Chlorogenic acid (5-caffeoylquinic acid: 5-CQA), neo-chlorogenic acid (3-caffeoylquinic acid: 3-CQA), and krypto-chlorogenic acid (4-caffeoylquinic acid: 4-CQA) were supplied by Sigma (Sigma Chemical Co., St. Louis, MO, USA). Anhydrous sodium carbonate was purchased from Fisher Scientific Company (Illkirch-Graffenstaden, France). 2,2-Azotis-(2-amidinopropane)-dilhydrochloride (AAPH), fluorescein (FL; 3′,6′-dihydroxyspiro[isobenzofuran-1[3H],9′[9H]-xanthen]-3-one or fluorescein sodium salt) were supplied by Wako Chemicals USA, Inc. (Richmond, VA, USA).

All solvents, reagents, and standards used in this work were HPLC grades (purity P 99%). All reagents and standard solutions were prepared with water purified on Veolia Water Systems™ (Veolia, Vendin-le-Vieil, France) S15 reverse osmosis apparatus.

### 3.2. Potato Samples

Potato samples were obtained from “Roquette Frères” Company, Lestrem, France. These samples are collected from different flow process steps: coarse or seed potato (CPo), lyophilized fresh potato peels (LFPPe), dehydrated potato pulp (DPP), potato peels under pellets form (PPP), ground and sieved pellets (GSP), lyophilized flesh potato (LFPo), red liquid (RL), and lyophilized red liquid (LRL). All these samples are shown in Figure 6.

### 3.3. Potato Samples; Preparation of Potato Samples and Extraction

The potato peels under pellet form (PPP) were ground to powder and sieved using two standard sieves (160 and 250 µm) to maximize extraction yields and lead to ground sieved pellets (GSP). In addition, dehydrated potato pulp (DPP) and red liquid (RL) were both used without any pretreatment.

In a typical experiment, ground potato peels or lyophilized LFPo or LRL (30 g) were suspended in 300 mL of solvent (methanol–water or ethanol–water 4:1 *v*/*v*). In the case of conventional extraction by maceration, the mixtures were incubated for 48 h at room temperature with continuous shaking. The solution was recovered by a two-step process consisting in a vacuum filtration (Whatman No. 1. filter paper) followed by centrifugation at 4000 rpm for 20 min at ambient temperature.

To study the influence of temperature on extraction, each potato by-products type (30 g) was heat-extracted under shaking for 30 min with 300 mL of 80% aqueous alcoholic solution, i.e., at 40 °C when using MeOH and at 75 °C in the case of EtOH. After cooling, suspensions were centrifuged at 4000 rpm for 20 min at room temperature. Supernatants were filtrated through Wathman N.42 filter paper for immediate HPLC analysis.

The ultrasound-assisted extraction procedure was used for the extraction of PPP with the appropriate solvent. Thus, 300 mL of solvent were added to 30 g of PPP; the mixture was sonicated in an ultrasonic bath for 15 min. The extract was centrifuged at 4000 rpm for 20 min at room temperature and was then subject to analysis.

The supernatants were collected and used for total phenolic compounds determination, HPLC analyses to determine the amount of chlorogenic acids in the extracts, and their free radical scavenging activity by ORAC tests.

### 3.4. Determination of Total Phenolic Compounds

Folin–Ciocalteu reagent is widely used to measure total phenolic compound amount in plant materials. For our purpose, the methods described by [23] and [34] were adjusted and undertaken with 0.5 mL of potato sample extract. Folin reagent (1 mL) was added to the extract, followed by 1 mL of 7.5% sodium carbonate solution (5% *w*/*w*). After shaking, the mixture was left for 90 min in darkness at room temperature, so a greenish-blue color appeared. Then, absorbance was measured at 765 nm, using an Ultrospec 2000 UV–Visible spectrophotometer (Pharmacia biotech, Guyancourt, France). A calibration curve (Figure 7) was obtained by using chlorogenic acid (5-CQA) as the standard, and the total phenolic compounds of extracts were estimated as milligram equivalent of chlorogenic acid per gram of potato sample powder.

### 3.5. Determination of Phenolic Acids (Chlorogenic Acids) in Potato Extracts by HPLC

The freshly prepared extracts were filtered through a 0.45 µm Acrodisc syringe filter (Pall Corp., Saint-Germain-en-Laye, France) prior to high-performance liquid chromatography analyses. The HPLC (SPD-10A Shimadzu) used for our purpose is equipped with a tertiary pump, thermostated auto-sampler, and a diode array detector. Phenolic extracts were separated using a reverse phase C-18 HPLC Uptispere HDO Gemini-NX column (5 m, 250 mm × 4.6 mm; Phenomenex, Inc., Torrance, CA, USA) equipped with a 4.6 mm × 2.0 mm guard column.

A flow rate of à.8 mL.min^−1^ was applied, and the elution program was defined as follows, considering the mobile phase as a binary solvent composed of 2 mM aqueous phosphoric acid (pH 3.5; solvent A) and methanol (HPLC grade, solvent B): 0–26 min at 25% of B, 26–51 min from 25% to 100% B, 51–56 min at 100% of B, 56–65 min from 100% to 25% of B. The injection volume was fixed at 20 µL, and UV detection was performed at 328 nm. Samples were analyzed in duplicates, and results are the mean of these two analyses.

Commercial chlorogenic acid (5-CQA), neo-chlorogenic acid (3-CQA), and crypto-chlorogenic acid (4-CQA) were also analyzed using the same HPLC operating conditions. The retention times of these 3 standards allowed us to clearly identify and quantify them in the extracts and also in extracts spiked with the standard components.

### 3.6. Oxygen Radical Absorbance Capacity (ORAC) According to the Catalog OxiSelect ™

#### 3.6.1. Reagent Preparation

Assay Diluent: The assay was diluted at 1/4 with deionized water. The solution was mixed to homogeneity and used for all sample and standard dilutions. This 1X Assay Diluent was stored at 4 °C.

Fluorescein probe was diluted at 1/100 with 1X Assay Diluent and mix to homogeneity to obtain 1X Fluorescein Solution.

Free Radical Initiator Solution AAPH was also prepared. Free Radical Initiator Solution in 1X PBS was freshly prepared for a concentration of 80 mg/mL. In a typical experiment, 160 mg of Free Radical Initiator powder was weighed in a conical tube, and the powder was dissolved with 2 mL of 1X PBS and mixed until homogeneity. Free Radical Initiator Solution is not stable and has to be used immediately.

#### 3.6.2. Preparation of Standards

Fresh standards were obtained by diluting the 5 mM Antioxidant Standard stock solution (Trolox) to 0.2 mM in Assay Diluent (Example: Add 10 μL of Antioxidant Standard stock tube to 240 μL of Assay Diluent).

A series of the remaining antioxidant standards was prepared in order to obtain different final concentration (5, 10, until 50 µM Trolox). A Trolox standard curve was generated following the same procedure as for the control solutions.

#### 3.6.3. Preparation of Samples

First, 25 μL of the diluted Antioxidant Standard (Trolox) or samples were added to the 96-well microtiter plate and 150 μL of the 1X Fluorescein Solution was added to each well with mix thoroughly. The plate was incubated for 30 min at room temperature.

Then, 25 μL of the Free Radical Initiator Solution (AAPH) were added into each well using either a multichannel pipette or a plate reader liquid handling system, and the reaction mixture was mixed thoroughly by pipetting to ensure homogeneity. The fluorometer cuvette holder was maintained at 37 °C during all measurements.

Reading samples and standard wells was immediately begun with a fluorescent microplate reader using an excitation wavelength of 480 nm and an emission wavelength of 520 nm. The wells were read with an increment between 1 and 5 min for a total of 60 min. Values obtained were used for calculation of results below.

Each Antioxidant Standard and sample was assayed in triplicate.

#### 3.6.4. Calculation of Results

Areas under the curves (AUC) were calculated using Excel Software. The AUC can be calculated from the equation below:(1)$$AUC=1+\frac{RFU_{1}}{RFU_{0}}+\frac{RFU_{2}}{RFU_{0}}+\ldots +\frac{RFU_{60}}{RFU_{0}}$$
where *RFU*_0_ = relative fluorescence value of time point zero, *RFU_x_* = relative fluorescence value of time points (e.g., *RFU*_2_ is relative fluorescence value at minute two).

The Net AUC was calculated by subtracting the Blank AUC from the AUC of each sample and standard.
Net AUC = AUC (Antioxidant) − AUC (blank)(2)

Therefore, the μMole Trolox™ equivalents (TE) of the unknown sample was calculated by comparing the standard curve of the Net AUC on the *y*-axis against the Trolox™ Antioxidant Standard concentration on the *x*-axis. Results (ORAC value) are expressed as TE per L or g of sample.

## 4. Conclusions

A lot of vegetables, fruits, nuts, and cereals already studied were found to contain either high or moderate amounts of phenolic acids. Our work was the first to determine the phenolic content (TPC and chlorogenic acids) and assess the antioxidant activity of starch potato by-products, including pellets, fresh peels, flesh, and red waters.

The total phenolic content was assessed using spectrophotometric analysis. Potato peel extracts were also analyzed using HPLC to identify the types of phenolic compounds. The total phenolic compounds (TPC) were extracted from potato by-product using hydroalcoholic solvents (methanol, ethanol) and two extraction methods (solvent maceration and heat-assisted extraction). TPC and CQA contents were quantified according to the Folin–Ciocalteu procedure and HPLC, respectively.

The highest content of TPC, i.e., 57mg/g fresh weight, was found in raw and lyophilized red liquids when using ethanol as a solvent. The heat-assisted extraction process increased the extraction yield. Under specific extraction conditions, unpeeled potato samples exhibit a higher TPC than those peeled, showing that TPC is accumulated in skin tissue.

Both chromatographic and spectrophotometric analysis showed that red liquid, lyophilized fresh peels, and pellets of potatoes do contain the highest levels of phenolics. They also had a highest ORAC values. These properties are dependent on the solvent and the method of extraction as well. The greatest amount of CQA, mainly the 5-CQA (around of 870 ± 39.7 μg/g), was obtained in the pellets and lyophilized fresh peels (skin vs. flesh). In addition, the greatest amounts of CQA were obtained with potato peels when using methanol as a solvent. Moreover, the heat-assisted extraction improved the 5-CQA concentrations from the potato peel.

The ORAC values varied between 1500 and 1650 μM TE/g, whatever the potato fractions. The results were higher than those of some fruits, vegetables, nuts, cereals, and sweet potato cultivar reported in the literature. The degree of correlation between TPC, CQA determination, and ORAC capacity indicate that the TPC can be used as an indicator of antioxidant evaluation of potato by-products and would be an easy tool to qualify enriched potato fractions for human or animal food and nutrition purposes.

ORAC and TPC were highly correlated, but ORAC and chlorogenic acid were moderately correlated and could be used during selection to improve quality. Based on these results, starch potato processing side products can be considered as natural sources high in active healthy micronutrients such as polyphenolic compounds with antioxidant properties. Future research studies are needed to study seasonal variations, stability, and the human health effects of these fractions.

## Figures and Tables

**Figure 1 molecules-26-00177-f001:**
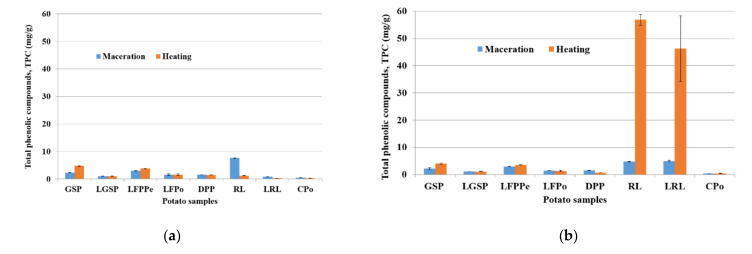
Effect of two extraction methods on potato by-products at on TPC content, (**a**) MeOH: H_2_O; 4:1 and (**b**) EtOH: H_2_O; 4:1 extracts.

**Figure 2 molecules-26-00177-f002:**
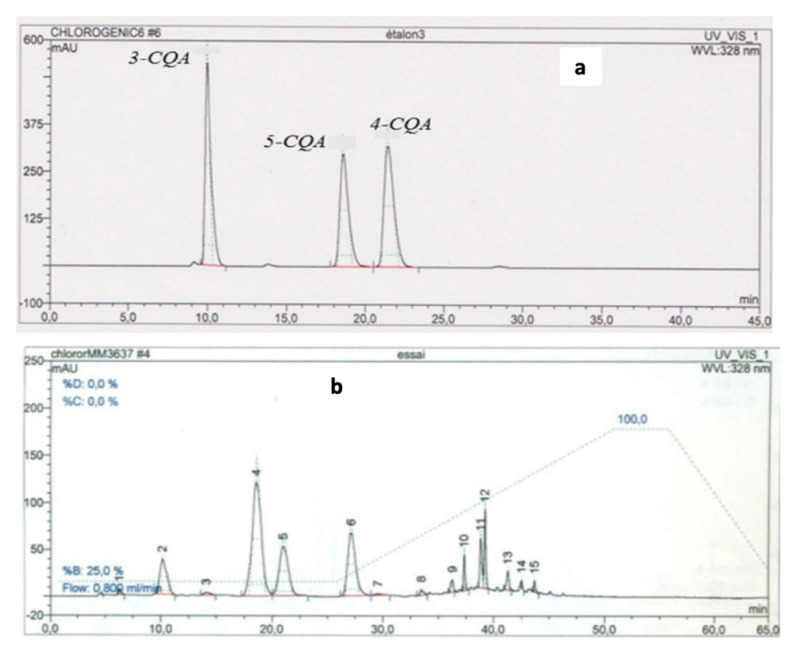
Gradient HPLC chromatograms of chlorogenic derivative acid standards (**a**) and phenolic acids on the methanol extract from the GSP fraction of potato samples (**b**). Stationary phase: column C18 Uptisphere HDO 120Å, 5 µm, 250 × 4 mm. Mobile phase: Solvent A: Phosphoric Acid 2 mM. Solvant B: Methanol. Flow: 0.8 mL/min. Gradient conditions: 0–26 min 25% of A, 26–51 min 25–100% of B, 51–56 min 100% of B, 56–65 min 100–25% of B.

**Figure 3 molecules-26-00177-f003:**
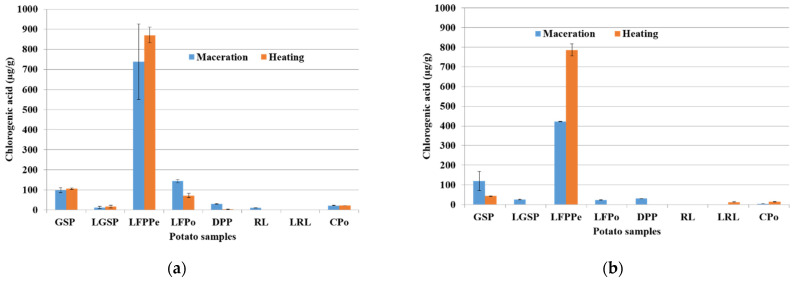
Effect of potato co-products at two extraction methods on chlorogenic acid content, (**a**) MeOH: H_2_O; 4:1 and (**b**) EtOH: H_2_O; 4:1 extracts.

**Figure 4 molecules-26-00177-f004:**
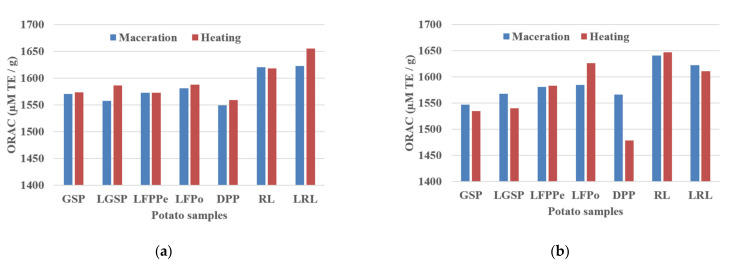
Variation of total oxygen radical absorbance capacity (ORAC) of potato samples under conditions of (**a**) MeOH: H_2_O; 4:1 and (**b**) EtOH: H_2_O; 4:1 extracts.

**Figure 5 molecules-26-00177-f005:**
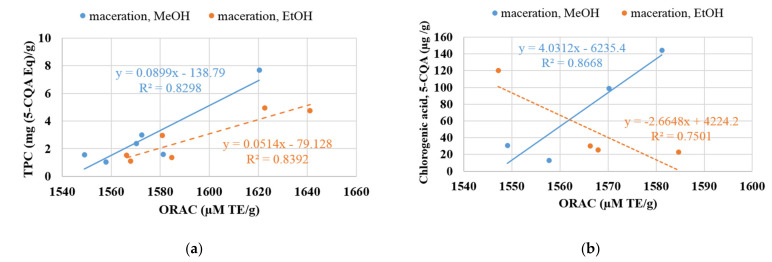
Correlation between total ORAC values and TPC contents (**a**) and chlorogenic acid (**b**) under maceration extraction with two solvents.

**Figure 6 molecules-26-00177-f006:**
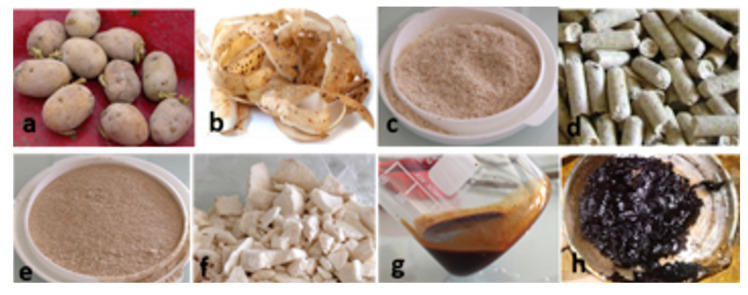
Different samples of potato by-products used in the present work: (**a**) coarse or seed potato (CPo), (**b**) fresh potato peels (FPPe), (**c**) dehydrated potato pulp (DPP), (**d**) coarse potato pulp or “pellets” (PPP), (**e**) ground and sieved pellets (GSP), (**f**) lyophilized flesh potato (LFPo), (**g**) red liquid (RL), and (**h**) lyophilized red liquid (LRL).

**Figure 7 molecules-26-00177-f007:**
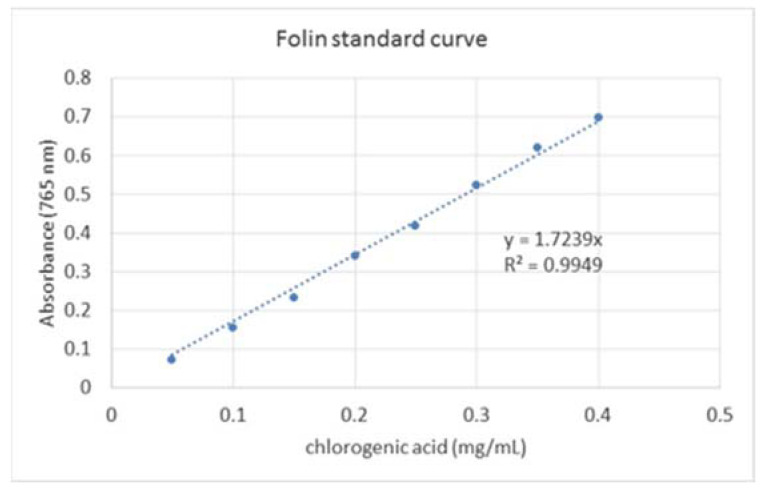
Calibration curve of chlorogenic acid (5-CQA) used for spectrophotometric analyses.

**Table 1 molecules-26-00177-t001:** Comparison of the methanol extraction methods of ground and sieved pellets (GSP) sample.

Methods of Extraction	Total Phenolic Compounds (TPC) by Colorimetric Method Folin–Ciocalteu Reagent) (mg (Chlorogenic Acid)/g of Sample	Individual Phenolic Compounds by HPLC (µg (Chlorogenic Acid)/g of Sample)
Heating at 40 °C/30 min	4.78 ± 0.23	106.45 ± 3.55
Maceration at RT/48 h	2.39 ± 0.06	102.64 ± 7.06
Soxhlet	3.17 ± 0.22	48.91 ± 3.38
Reflux	2.07 ± 0.09	83.20 ± 1.26
Percolation	1.53 ± 0.18	31.94 ± 4.39
Ultrasound	1.67 ± 0.09	94.84 ± 5.40

## Data Availability

The data presented in this study are available in article.

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
