# Peer review of "Potato By-Products as a Source of Natural Chlorogenic Acids and Phenolic Compounds: Extraction, Characterization, and Antioxidant Capacity"

_molecules, 2020, doi:10.3390/molecules26010177_

Round 1

Reviewer 1 Report

Overview:

In this study the authors examine the potential utilization of potato co-products industrially produced by Roquettes-Freres to obtain anti-oxidant extracts.

The authors analyze the total phenolic compounds (TPC) and chlorogenic acid content of potato by-product extracts using two different hydro-alcoholic solvents (methanol, ethanol) and two extraction methods (maceration and heating assisted-extraction). Furthermore, they evaluate the antioxidant activity of potato by-product extracts using the total Oxygen Radical Absorbance Capacity (ORAC) tests.

They found the highest content of TPC, i.e. 57mg/g fresh weight, in raw and lyophilized red liquid when using ethanol as a solvent. TPC seems to be accumulated in skin tissue. On the other hand, the highest amount of CQA (around of 870 μg/g), was obtained from pellets and lyophilized fresh peels (skin vs flesh) using methanol as a solvent.

The ORAC values (1500 - 1650 μM TE/g) are higher than those of some fruits, vegetables, nuts, cereals and sweet potato cultivar.

The authors found that TPC can be used as an indicator of antioxidant evaluation of potato by-products and would be an easy tool for qualifying enriched potato fractions for Human or animal food and nutrition purposes.

The paper is overall well written and much of it is well described. Research is interesting and the data provided are well discussed.

2.1. Major comments:

1) In the results section, figure 1a, b and figure 3 a, b how many times the extractions from the different parts of the starch potato have been performed? Furthermore, have been done a statistical analysis? If yes, please, it is better to report it in the graph and to describe the statistical analysis method used in the material and methods section.

2) In the results section, figure 4a, b how many times, for each sample, have been determined ORAC values? In the graph there aren’t error bars. Why? Please, report it in the graph and describe the statistical analysis method used in the material and methods section.

3) How Do you explain that the highest content of TCP was obtained with ethanol as solvent while the highest content of Chlorogenic acid was obtained with methanol? Please better discuss this point in the text, it’s not very clear.

Author Response

The authors apologize to the reviewer for the delay in responding, but the period is difficult with the health crisis. The authors thank the reviewer for taking the time to review and comment on our proposal.

We will respond separately to the 3 major points expressed by the reviewer.

1) In the results section, Figure 1a, b and Figure 3a, b the extractions on the different parts of the starch potato were performed twice (duplicate). There was no statistical analysis to perform during our study.

2) In the results section, figure 4a, b for each sample a single measurement of the ORAC values ​​was carried out.

3) We have completed our explanation in the text of the attached revised manuscript (line 216-222). To provide an explanation in our response to the reviewer in addition to the one written in the revised version: This is explained by the selectivity of the extraction with methanol, which allows a better extraction of CQA selectively, while ethanol allows a more global extraction of polyphenols. Moreover, in the fractionation of biomass, one always proceeds to the extraction of tannins and other extractables by an EtOH / H2O mixture, for a better efficiency.

Reviewer 2 Report

The manuscript: molecules-835846, title: “Potato by-products as a source of natural chlorogenic acids and phenolic compounds: extraction, characterization and antioxidant capacity” submitted for review is very interesting and at a high scientific and technical level. Describes important phenomena of chlorogenic acid abundant and extraction from potato and potato-rich products.

First section is written quite well, but I think literature review would be better if Authors use those two publication about concentration of chlorogenic acid in potatoes:

Kazimierczak R., Średnicka- ober D., Hallmann E., Kopczynska K., Zarzynska K.: The impact of organic vs. conventional agricultural practices on selected quality features of eight potato cultivars, Agronomy, 9, 2019, 1-15;

Vaitkeviciene N., Kulaitiené J., Jariene E., Levickiene D., Danillčenko H., Średnicka - Tober D., Rembiałkowska E., Hallmann E.: Characterization of bioactive compounds in colored potato (Solanum Tuberosum L.) cultivars grown with conventional, organic, and biodynamic methods, Sustainability,  12, 7, 2020,  1-13.

Page 9, line 336: how many potatoes were in examined sample – please add this information to relevant part of manuscript.

Page 7 Figures 3- 4: there is no statistical tools on the figures. I’m interested in  If the Authors use statistical analysis of obtained results In Material and Method section is no any description of that. – Could you explain it and add relevant information to section.

Results and discussion section is written quite well.

Conclusions are formatted correctly.

I think the work is interesting and valuable and should be published after minor corrections.

Author Response

The authors apologize to the reviewer for the delay in responding, but the period is difficult with the health crisis. The authors thank the reviewer for taking the time to review and comment on our proposal.

We will respond separately to the 3 major points expressed by the reviewer.

1) The two publications suggested by the reviewer, on the concentration of chlorogenic acid in potatoes, have been added to the revised version of the attached manuscript (line 97-98).

2) It is difficult to estimate how many potatoes were initially in the sample examined. The samples were collected throughout the industrial process of the Roquette Company, knowing that each batch treats more than 5 tonnes of raw materials.

3) Page 7; In the results section, Figure 3a, b the extractions on the different parts of the starch potato were performed twice (duplicate). There was no statistical analysis to perform during our study. In the results section, figure 4a, b for each sample a single measurement of the ORAC values ​​was carried out.